# Enhanced Operational Flexibility of a Small Run-of-River Hydropower Plant

**Jean Decaix** [1,*] , **Anthony Gaspoz** [2] , **Vlad Hasmatuchi** [2] , **Matthieu Dreyer** [3] , **Christophe Nicolet** [3] , **Steve Crettenand** [4] and **Cécile Münch-Alligné** [1,2]

1   Institute of Sustainable Energy, School of Engineering, HES-SO Valais-Wallis, Rawil 47, 1950 Sion, Switzerland; cecile.muench@hevs.ch
2   Institute of Systems Engineering, School of Engineering, HES-SO Valais-Wallis Rawil 47, 1950 Sion, Switzerland; anthony.gaspoz@hevs.ch (A.G.); vlad.hasmatuchi@hevs.ch (V.H.)
3   Power Vision Engineering Sàrl, Rue des Jordils 40, CH-1025 St-Sulpice, Switzerland; matthieu.dreyer@powervision-eng.ch (M.D.); christophe.nicolet@powervision-eng.ch (C.N.)
4   FMV SA, Rue de la Dixence 9, CP 506, CH-1951 Sion, Switzerland; Steve.Crettenand@fmv.ch
*   Correspondence: jean.decaix@hevs.ch

**Abstract:** Over the last two decades, the public policies for promoting new renewable energies allowed the growth of such energies around the world. Due to their success, the policies are changing, forcing the producers to adapt their strategy. For instance, in Switzerland, the feed-in tariff system has been modified in 2018 to promote an electricity production from renewable energies that matches the demand. For small hydraulic power plants owners, such a change requires to increase the flexibility of their fleet. The SmallFLEX project, led by HES-SO Valais, aims at demonstrating on the pilot site of Gletsch-Oberwald owned by Forces Motrices Valaisannes SA, the possibilities to increase the flexibility of the power plant and to provide new services. The paper focuses on the methodology followed to warranty the use of the settling basin, the forebay tank, and the third upper part of the headrace tunnel as a new smart storage volume. By combining laboratory tests, numerical simulations, and on-site measurements, the new range of operating conditions has been defined. These data can be used to foresee economic gains. The methodology and the outputs of the project can be useful for performing such a study on other power plants.

**Keywords:** flexibility; hydropower; pilot demonstration



## 1. Introduction

In the last three decades, the development of new renewable energies have been encouraged by public policies all around the world leading to average annual growth rates of 37% for solar photovoltaic and 23% for wind energy [1]. However, these stochastic sources of energy require additional flexible production to ensure the production-demand balance on the electricity grid [2]. Switzerland follows this trend with its 2050 Swiss Energy Strategy that aims at developing and increasing the part of renewable energies in the energy mix and in the meantime at stopping nuclear power plants [3]. Flexibility can be achieved by using hydropower plants that have storage capacity allowing to shift in time the production of electricity. Such flexibility is also encouraged by the feed-in tariff in Switzerland [3] that forces the producers to match the demand.

Regarding hydropower plants, additional flexibility can be achieved by increasing the height of dams or the use of future proglacial lakes [4–6], developing and increasing the pumped storage capacities [7,8], adding frequency converters [9], developing battery hybridization [10,11], implementing a hydraulic short circuit [12], or by using new storage capacities [13,14]. The last solution can increase the flexibility of small run-of-river power plants without too high environmental and economic costs.

In the framework of the SmallFlex project [15], the team led by Haute-Ecole Spécialisée de Suisse Occidentale (HES-SO) in collaboration with Power Vision Engineering (PVE),

Forces Motrices Valaisannes SA (FMV SA), the Swiss Federal Institute for Forest, Snow and Landscape Research (WSL), the Swiss Federal Institute of Aquatic Science and Technology (EAWAG), and the Swiss Federal Institute of Technology Lausanne (EPFL), focuses on the Gletsch-Oberwald (KWGO) small hydro power plant owned by FMV SA. This run-of-river power plant equipped with two Pelton turbines has a 2.1-km long head race tunnel, part of which can be used as new storage capacity [16]. The Smallflex project aims at demonstrating on a pilot site the possibility to increase production flexibility while limiting new additional costs and minimizing the impact on the environmental [17].

The present paper focuses only on the technical assessments of the new flexibility offered by emptying the settling basin, the forebay tank, and the third upper part of the head race tunnel. Firstly, the main information regarding the power plant and the input data are described. Then, the upstream investigations carried out before the field tests are detailed and followed by the results of the on-site measurements. Finally, the main findings are summarized in the conclusion with an opening on the economic and environmental values of the results.

## 2. Description of the Power Plant

The small run-of-river hydro power of KWGO owned by FMV is located close to the sources of the Rhone river in the canton of Valais in Switzerland, see Figure 1 and Novotny [18]. The hydrology flow regime is typical of the Alpine regions with a low flow discharge during winter since most of the precipitations fall in the form of snow, whereas during spring and summer, the discharge reaches its highest values due to snow and glacier melt. Therefore, the power plant is equipped with two Pelton turbines of 7.5 MW each. The nominal head is equal to 287 m and the power plant flow discharge ranges from $0.145 \text{ m}^3 \text{ s}^{-1}$ to $5.8 \text{ m}^3 \text{ s}^{-1}$, which corresponds to a specific speed $n_q = 0.1 \text{ s}^{-1}$. The intake consists of a settling basin and a forebay tank with a storage capacity of $2500 \text{ m}^3$.

The large range of flow discharges under which the Pelton turbines can operate is obviously an advantage for flexibility. Therefore, the first task of the Smallflex project was dedicated to the detailed analysis of possible unused or new storage volumes [16]. This study allows first on deciding to drill two passages between the settling basin and the forebay tank. These two passages are equipped with gates that are closed from May to October, when the flow discharge and the sediment content are high. Between November and April, the gates are opened since the sediment content is low and the settling basin feature is not necessary. Moreover, the study shows that the use of the third upper part of the head race tunnel should be interesting regarding both the available volume ($4650 \text{ m}^3$) and the low cost required to use it, since there is no need for construction or excavation.

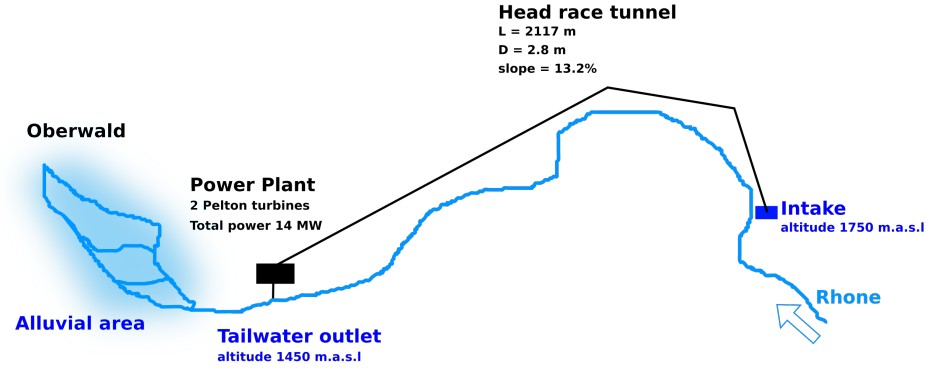

**Figure 1.** Schematic representation of the KWGO power plant layout.

## 3. Technical Investigations before On-Site Tests

Despite the amount of this additional volume and its rather cheap availability, it cannot be used without performing several technical investigations and analyses. Indeed,

the use of this volume requires to empty and to fill the upper part of the head race tunnel several times per year, which can provoke damages of the tunnel by air entrainment [19], compression, and tensile stresses. Both risks are critical for the Pelton turbines as they can cause irreparable damages due to the blockage of the flow in the nozzle by small rocks or due to air bubble expansion in the jet. Furthermore, the Pelton turbines will be operated at a lower head than the one for which they are designed. Therefore, their efficiency will first decrease slowly until reaching a head at which an abrupt drop will occur due to the so-called "falaise" effect [20,21]. Finally, the capacity of the power plant to provide ancillary services respecting the Swissgrid requirements must be considered. To tackle all these points, four different studies have been carried out:

1.  For the risk of erosion, an external expert has been appointed to certify no risk at least among several cycles of emptying and filling operation. This point is no more discussed in the paper;
2.  For the risk of air entrainment, a reduced scale model of the forebay tank and the head race tunnel has been built at the HES-SO Valais hydroelectricity laboratory;
3.  For the efficiency drop due to head lowering, analytical calculations and an analysis of the digital twin database have been carried out;
4.  For the primary control services, a 1D dynamic model of the plant was made using the SIMSEN software. This model, whose parameters could be calibrated during the commissioning tests, was also used as a basis for the real-time monitoring of the power plant with the digital twin Hydro-Clone®, see [22,23].

*3.1. Investigation of the Risk of Air Entrainment*

During the emptying of the forebay tank and the upper part of the head race tunnel, two different processes can be responsible of air entrainment: The formation of a vortex at the free surface in the forebay tank and the hydraulic jump that occurs in the head race tunnel when its upper part is emptied.

Regarding the vortex-induced air entrainment in the forebay tank, a formula was derived by Möller et al. [24] that links the critical relative intake submergence $(h/D)_{cr}$ (with $h$ the intake submergence and $h = 0$ at the intake pipe axis) to the intake Froude number $F_D$ (see Equation (1)):

$$\left(\frac{h}{D}\right)_{cr} = -2.5F_D^{-0.45} + 5.3 \text{ for } 0.26 \leq F_D \leq 1.2 \text{ and } 0.75 \leq (h/D)_{cr} \leq 3 \tag{1}$$

$$F_D = \frac{V_D}{\sqrt{gD}}.$$

With $D$ as the intake diameter and $V_D$ as the velocity of the water through the intake section. In the present study, the Froude number $F_D$ at the nominal operating conditions is equal to 0.142 that is smaller than the lower bound of 0.26 for which the formula is derived. However, by extrapolating the formula, the critical relative intake submergence $(h/D)_{cr}$ should be negative with a value of $-0.72$. Such a result suggests that no vortex should occur during the emptying of the forebay tank.

Regarding the risk of air entrainment by the hydraulic jump occurring in the head race tunnel after emptying the forebay tank, an analysis based on reference [19], partially updated in a discussion published in 2011 [25], discards this risk. Air bubbles are expected to be entrained by the flow downstream the hydraulic jump if the ratio between the dimensionless flow discharge $q^* = Q^2/(gD^5)$ and the slope $S$ of the pipe is larger than 0.83. For the present study, the slope $S$ of the head race tunnel is equal to 0.13 and for the maximum discharge of the power plant, the dimensionless discharge $q^*$ equals to 0.012, which gives a ratio $S/q^*$ of 10.8 largely above 0.83.

Since no formula in the literature are available for the flow condition considered in this study, a reduced scale (1:20) model (see Figure 2) has been built to investigate the risk of air entrainment. The test rig is designed to perform Froude similitude tests. However, scale

effects regarding the volume of air entrained by a vortex in the forebay tank are expected since at a model scale and for the nominal operating condition, the Reynolds number $R_D$ is equal to $2.5 \times 10^4$, which is lower than the threshold of $6 \times 10^5$ prescribed in [24]. However, if a vortex forms at the free surface, it should be visible at the model scale.

Two forebay tanks have been considered to explore both the summer mode (gates closed between the settling basin and the forebay tank) and the winter mode (gates opened).

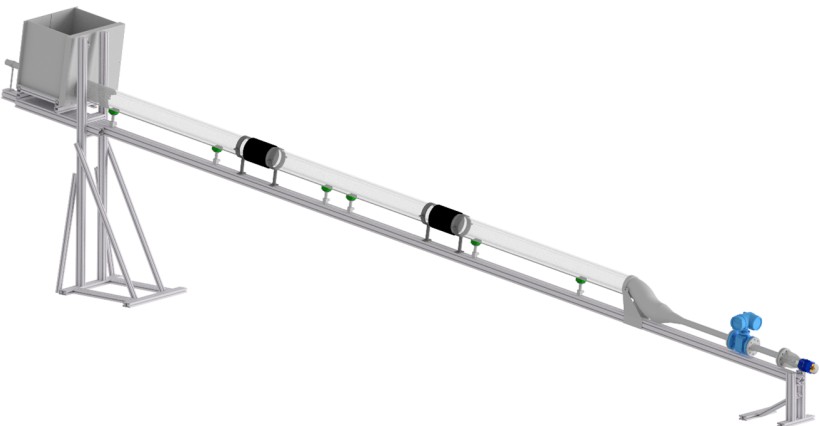

**Figure 2.** 3D CAD view of the reduced scale model (scale 1:20) used for the investigation of the air entrainment during the emptying of the forebay tank and the head race tunnel.

A high speed video camera has been used to visualize the entrainment of air bubbles in the pipe or the presence of air vortex in the forebay tank. At the maximum flow rate, no air entrainment or vortex are observed neither in the forebay tank nor in the partially emptied pipe (see Figure 3).

When the pipe is partially emptied, a hydraulic jump occurs leading to the formation of a mixture of air and water downstream see Figure 3b). However, it is observed that buoyancy forces are enough to allow the bubbles to reach the free surface before being captured by the flow downstream the hydraulic jump. By increasing the flow rate to 150% of the maximum flow rate, air bubbles are entrained downstream and a vortex appears in the forebay tank. Therefore, the model tests conclude that air entrainment is unlikely at the flow rate operating conditions of the Hydraulic Power Plant (HPP).

b)

a)

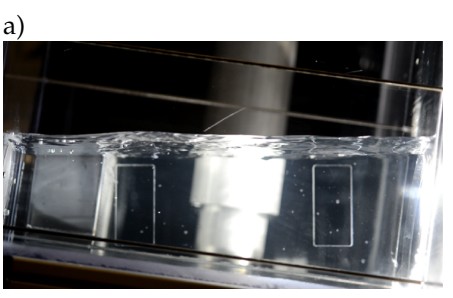
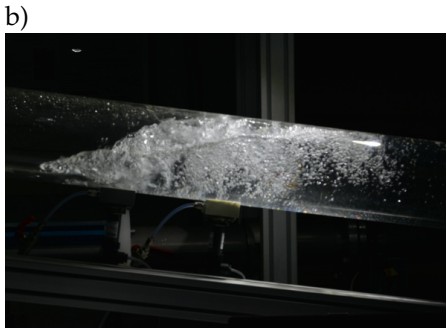

**Figure 3.** (**a**) Instantaneous picture of the free surface in the reduced scale forebay tank. (**b**) Instantaneous picture of the free surface in a partially emptied pipe. The flow discharge is set to its maximal value.

### 3.2. Efficiency Drop and "Falaise" Effect

By lowering the head, the jet velocity decreases. Below a critical value, the jet velocity is too low and water can not reach the runner bucket. Consequently, no torque is transmitted to the runner and the output power drops down to zero. By carefully investigating the flow in the bucket, the drop in power begins when the jet velocity prevents the water sheet inside

a runner bucket to leave before the bucket is impinged by a second jet [20]. Therefore, a shock occurs between the water sheet and the jet, which leads to an efficiency drop called the "falaise" effect. The prediction of the occurrence of this effect can be deduced by various methodology: Analytical calculations, data base analyses, or computational fluid dynamics.

Analytical calculations can give only an approximate value of the head at which the "falaise" effect should appear [21]. This head is linked to a critical value of the peripheral speed coefficient $k_m$ expressed by:

$$k_m = \frac{U_m}{\sqrt{2gH}} \tag{2}$$

with $U_m$ as the peripheral speed of the Pelton wheel on the jet runner circle $R_m$, $g$ is the gravitation acceleration, and $H$ is the head. At the nominal operating condition, the value of $k_m$ usually ranges between 0.45 and 0.48. For the KWGO power plant, the nominal $k_m$ is equal to 0.48.

The critical peripheral speed coefficient $k_{m,cr}$ can be estimated using Equation (3) for a jet without thickness:

$$k_{m,cr} = \frac{\alpha_0 - \pi/N}{\tan \alpha_0}. \tag{3}$$

With $N$ as the number of buckets and $\alpha_0$ as the characteristic bucket position angle. This angle can be computed by solving the implicit Equation (4) that links $\alpha_0$ to $k_m$.

$$k_m = \frac{2\pi}{N} \frac{2\lambda - 1}{\tan \alpha_0 - \tan(\alpha_0 - 4\lambda\pi/N)} \tag{4}$$

where $\lambda$ is usually set to 1.05 in order to consider real operating conditions.

For $N = 21$ and $k_m = 0.48$, the angle $\alpha_0$ equals $30°$ and $k_{m,cr}$ equals 0.65. However, the critical speed coefficient $k_{m,cr}$ depends on the jet layer $y$ as shown in Figure 4 derived from [21]. Therefore, the lowest value of the $k_{m,cr}$ is around 0.55 gained for $y/d_0 = 0.5$ and using Equation (2), the critical head is around 220 m. Below this head, the "falaise" effect should occur.

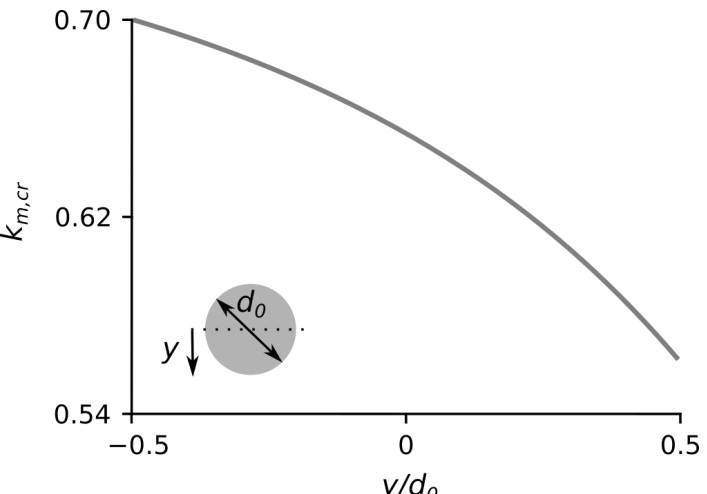

**Figure 4.** Critical peripheral speed coefficient $k_{m,cr}$ as a function of the dimensionless jet layer $y/d_0$ with $d_0$ as the jet diameter (see [21] for a detailed analysis).

Such a value has also been estimated using the extrapolated turbine characteristic of the digital twin as shown in Figure 5 where it is noticeable that the efficiency drops from approximately 0.8 for a net head of 220 m to less than 0.5 for a net head of 200 m.

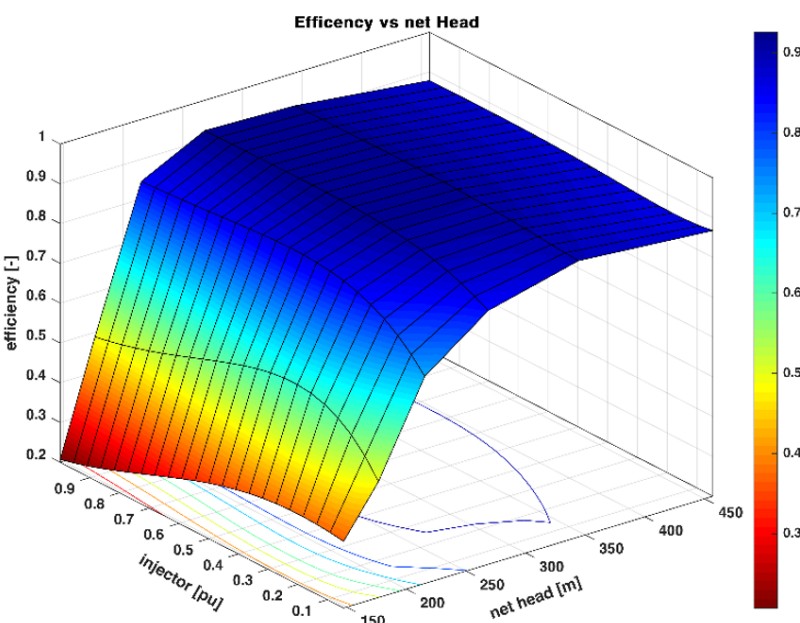

**Figure 5.** Estimation of the "falaise" occurrence based on the turbine characteristic used by the digital twin.

In addition, to anticipate the behavior of the power plant prior to on-site tests, simulations using the 1D dynamic model have been carried out including also an emergency shutdown when the head is below 190 m (see Figure 6). These simulations showed the possibility of performing an emergency shutdown during the dewatering of the headrace tunnel without major risk to the power plant in terms of pressure fluctuations induced by the water hammer.

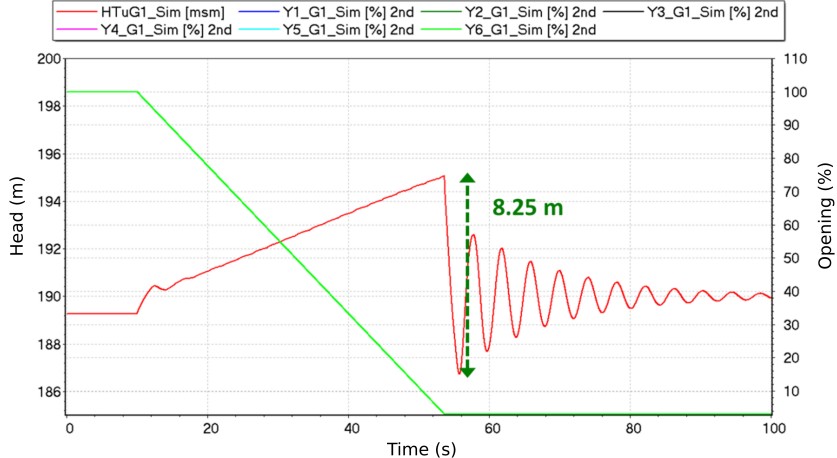

**Figure 6.** Simulation of an emergency shutdown at a low head around 189 m. $HTuG1\_Sim$ is the simulated head available for the group 1 and $Yi\_G1\_Sim$ is the needle stroke in % for each nozzle.

### 3.3. Investigation of Primary Control Services

Among the ancillary services, the primary control ensures that the balance between production and consumption is restored within seconds following a disturbance. This activation is carried out directly and automatically in the power plant using the turbine controllers. To be eligible to provide this service, the power plant must be pre-qualified by Swissgrid (the Swiss Transmission System Operator) to ensure that it is able to provide the primary control capacity, which must be kept in reserve at all times. The 1D numerical

model is thus used to simulate several primary control scenarios to disclose how much reserve power the power plant can provide, while meeting the Swissgrid qualification criteria. The dynamic limits of the power plant can thereby be determined, as well as the inlet and outlet water level variations induced by the provision of primary control.

According to Swissgrid's specifications, it must be possible to consistently activate the primary reserve power within 30 s and to deliver it for at least 15 min for any quasi-stationary frequency deviation of ±200 mHz. In addition, the power production must be within the tolerance bands shown in Figure 7. The amount of power delivered by a generating unit as a function of the frequency deviation in the network is typically characterized by the value of the permanent droop, defined as $BS = (\Delta f / f_{ref}) / (\Delta P / P_{ref})$. For instance, a permanent droop of $BS = 4\%$ corresponds to the provision of ±10% of the nominal power in case of a frequency deviation of ±200 mHz.

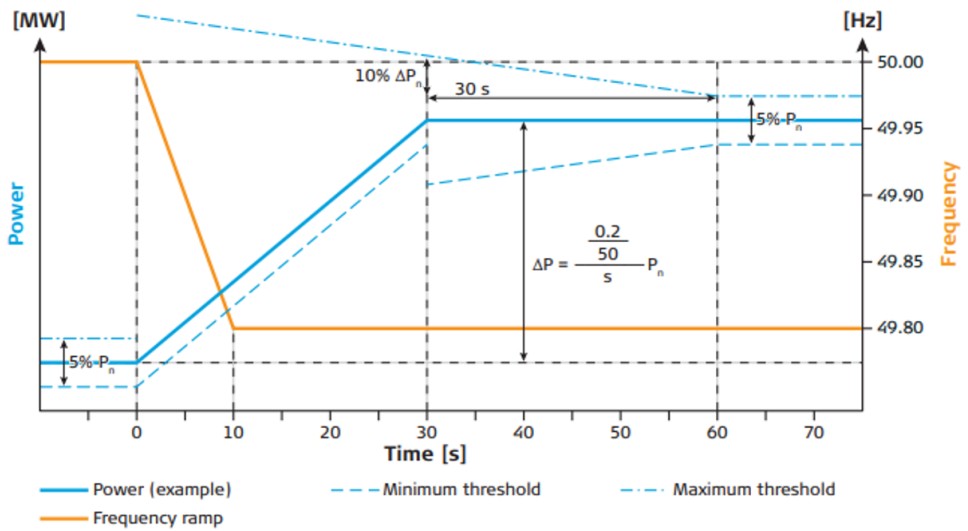

**Figure 7.** Tolerance band of the Swissgrid qualification [26].

Considering the available water volume in the settling basin, the capacity of the primary control is $\Delta P = 2 \times \pm 4$ MW, corresponding to a permanent droop of $BS = 0.755\%$. Regarding the Swissgrid qualification test in terms of frequency response, the capacity of the primary control is lowered to $\Delta P = 2 \times \pm 3$ MW, which corresponds to a permanent droop of $BS = 1\%$. However, the permanent droop acts as a gain on the power control loop regulator. Therefore for an active power set point change, a low permanent droop can lead to a more stable behavior than a high permanent droop value, although the corresponding contribution to the primary control is more important. Considering this stability criterion, the permanent droop value that would guarantee a stable operation for a set of PID parameters can be selected between $BS = 4\%$ and $BS = 6\%$. For a permanent droop of $BS = 4\%$, the primary control capacity is of $\Delta P = 2 \times \pm 0.75$ MW. Figure 8 displays the frequency response for a permanent droop of $BS = 4\%$ that matches the Swissgrid criteria.

By reproducing the hydraulic transients of the power plant, the simulations yield pressure fluctuations throughout the waterway, which can be post-processed to derive the level of stress on the penstock and evaluate the potential impact of primary control on the servie life of the penstock [27]. The pressure fluctuations resulting from the primary control have been estimated by simulating the response of the power plant to a realistic Swissgrid signal representative of the frequency variations in the grid. As shown in Figure 9, the pressure fluctuations induced by this ancillary service do not exceed 2.4% of the nominal head at the bottom of the penstock. In addition, the pressure fluctuations amplitude, along the entire penstock generated by the grid frequency control never exceed 3% of the design static pressure. Since these fluctuations are smaller than 5% of the design pressure, the

British Standard EN 13445-3 (EN BS 2009) suggests that they do not contribute to the fatigue of the pipe structure. It is therefore concluded that the impact of the primary control on the service life of the penstock is not significant.

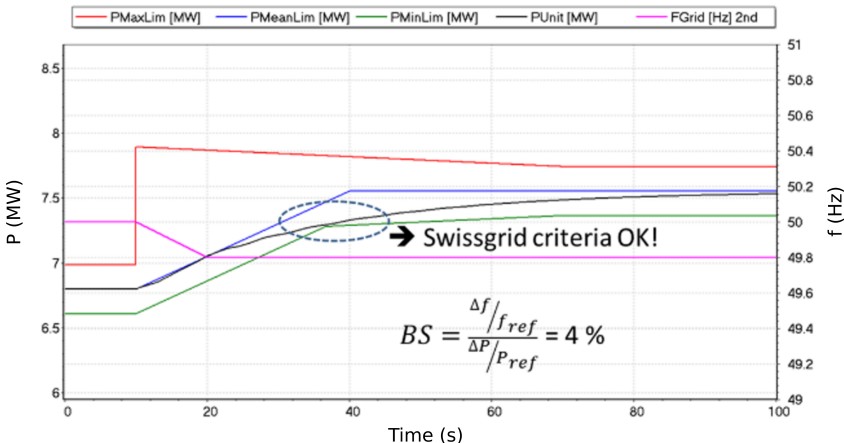

**Figure 8.** Simulated frequency response scenario meeting Swissgrid criteria with a permanent droop of $BS = 4\%$. *PMaxLim* is the limit of the maximum power accepted, *PMeanLim* is the limit of the mean power maximum accepted, *PMinLimit* is the limit of the minimal power accepted, *PUnit* is the power of the unit simulated, and *FGrid* is the frequency of the grid.

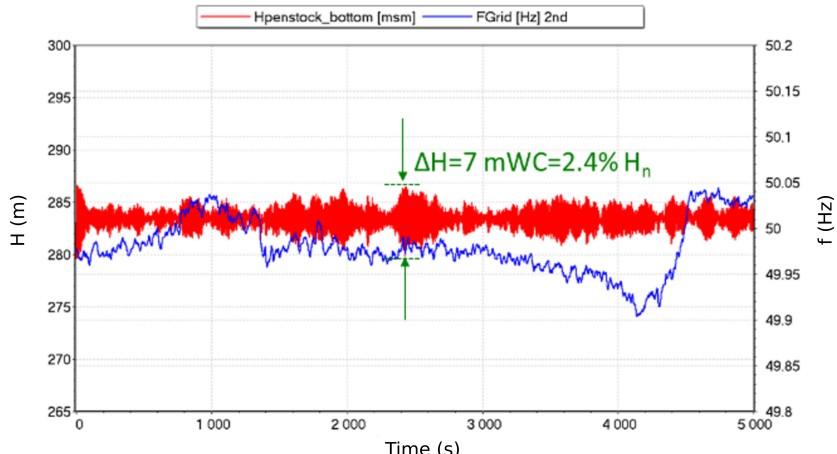

**Figure 9.** Simulation of the pressure fluctuations at the penstock bottom (*Hpenstock_bottom*) generated by the grid frequency control.

## 4. On-Site Tests

The on-site tests, focusing on power plant behavior, have been split in two campaigns that took place respectively in November 2018 for the first campaign and in May 2020 for the second campaign. During the first campaign, no emptying of the the head race tunnel was done and only the water level in the settling basin has been lowered because the Pelton turbines were still under warranty. This campaign allows validating the methodology developed to plan the production programs based on the available storage volume and the estimation of the incoming flow. It also allows for testing the instrumentation that consists of:

- One Kistler IEPE three-axis accelerometer (**Acc**) with a range of $\pm 10$ g;
- One GRAS IEPE microphone (**Mic$_T$**) with a range of $\pm 80$ Pa;
- One EH Proline Prosonic flow 93T flow-meter (**Q$_{u2}$**) with a range from 0 to 4 $m^3 s^{-1}$;
- One EH Cerabar PMP21 absolute pressure sensor (**P$_{u2}$**) with a range from 0 to 40 bar abs;
- One EH Cerabar T PMC131 absolute pressure sensor (**P$_{atm}$**) with a range from 0 to 1 bar abs;

- One Sick optical tachometer tachometer (**N**).

Figure 10 shows the location of each sensor around the Pelton turbine. The acquisition system is a National Instruments cDAQ 9174 with a dedicated Labview interface and is equipped with four cards:

- $1 \times$ NI 9233: $4 \times$ AI ±5 V (IEPE), 50 kS s$^{-1}$ ch$^{-1}$, 24 bits;
- $1 \times$ NI 9402: $4 \times$ LVTTL High-speed DIO 5V, 55 ns;
- $2 \times$ NI 9203: $8 \times$ AI ±20 mA, 200 KS s$^{-1}$, 16 bits.

The acquisition frequency is set to 10 kHz for the accelerometer, the microphone, and the tachometer. For the other sensors, the acquisition frequency is set to 200 Hz.

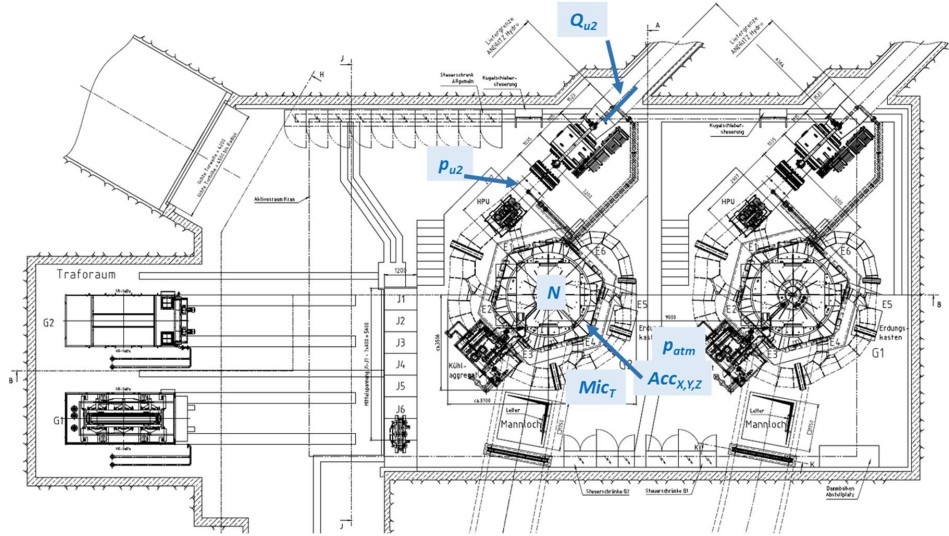

**Figure 10.** Locations of the various sensors around the Pelton turbine of group 2 with $Acc_{x,y,z}$ as the three axis accelerometer, $Mic_T$ as the microphone, $N$ as the tachometer, $P_{atm}$ as the pressure sensor in the power plant house, $P_{u2}$ as the pressure sensor set at the inlet of the manifold, and $Q_{u2}$ as the flow-meter.

An Infra-Red (IR) Axis M2025-LE network camera has been used to film the free surface in the forebay tank during the tests (see Figure 11).

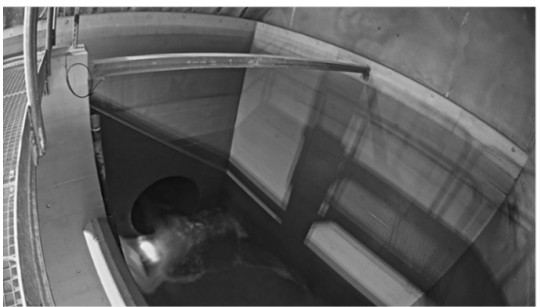
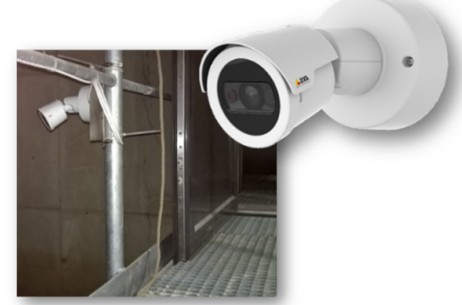

**Figure 11.** IR camera with an image of the forebay tank emptied.

Finally, the digital twin Hydro-Clone®, which allows for real-time monitoring of the power plant, is used to monitor the transient behavior of the KWGO power plant during the test campaigns and to identify possible malfunctions or extreme loads. Among the features provided by this digital twin, the most valuables for the on-site measurements are: The monitoring of the transient phenomena, the detection of significant pressure variations, the detection of anomalies (unexpected gate or valve closures, sensor failure, etc.), the monitoring of non-measurable quantities such as the pressure in the head race tunnel and the variation of the efficiency of the turbines and generators.

The second campaign has been designed to determine the performance map of the Pelton turbines and to define the safety limits of the power plant under low head operating conditions, i.e., by emptying the first third upper part of the head race tunnel. The design of the experiments requires to define specific maps allowing to provide a one-day production plan to FMV without losing water. One of these maps is shown in Figure 12 for an inflow discharge of 1.5 m$^3$ s$^{-1}$. It represents the available head as a function of the peak duration and power demand. The number of opened nozzles is also added on the map for information. The thick black line refers to the maximal possible power that can be satisfied during a peak of production. The yellow dash line shows the trajectory of a possible peak of production starting at a head of 284 m during 0.5 h at a constant power of 6.5 MW. When the head reaches a value of 270 m, the operating point is a switch to a constant power of 5.8 MW over 1 h. At the end of the peak, the head is equal to 220 m.

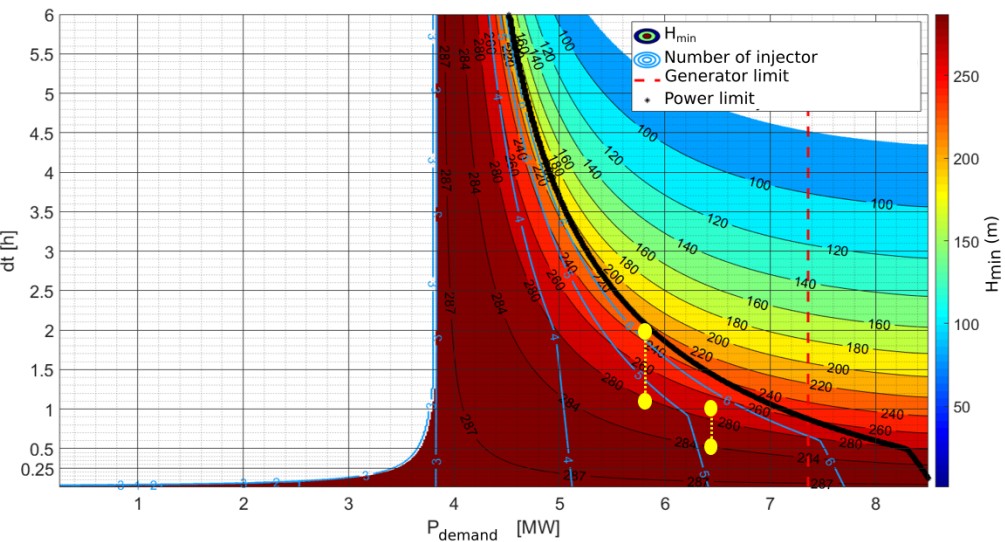

**Figure 12.** Contours of the available head as a function of the peak duration and the power demand. The thick black line refers to the maximal possible power for a given peak of duration. The yellow dash line represents the trajectory of a possible peak of production. The map is drawn for an inflow discharge at the intake of 1.5 m$^3$ s$^{-1}$.

## 5. Results

During the two days of tests, three peaks of production have been performed (see Figure 13). The first group G1 was used only to adjust the power of the entire power plant to avoid wasting water. Once on the first day and twice on the second day, the head has been lowered until a value around 190 m. The duration have been decreased from 3 h 35 for the first peak to 1 h 15 for the third one. The energy produced during the peaks is respectively of 12.8 MWh, 8.7 MWh, and 5.8 MWh. The power of the group has been varied during the peak (mainly the third peak) to sweep a large range of operating conditions.

Figure 14 gives more details of the third peak by displaying the time history of the head, the power, the efficiency, the flow discharge, and the number of opened injectors for group 2. In this figure, below a head of 250 m, it is noticeable that the efficiency is higher for low flow discharges. Interestingly, no oscillations of the head or the flow discharge are observed during the emptying.

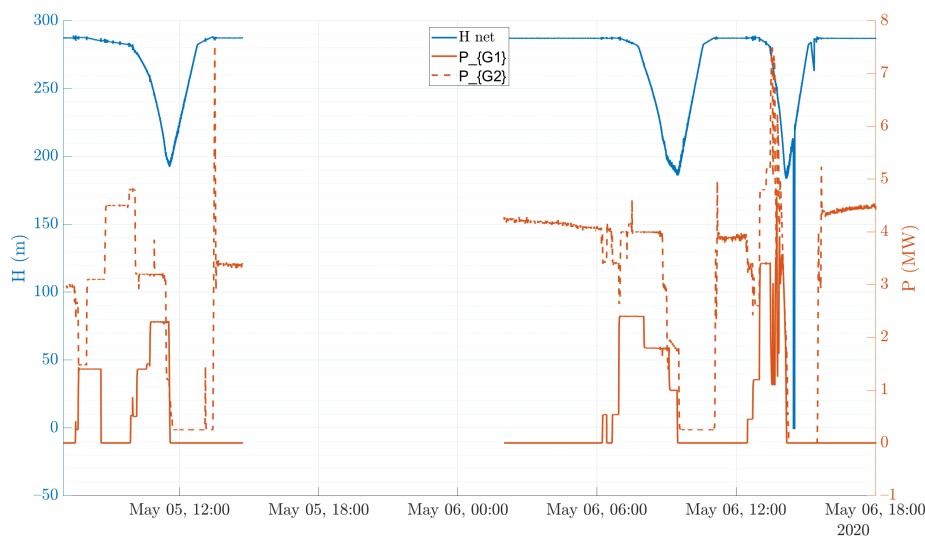

**Figure 13.** Time history of the head and the power of the two Pelton groups during the two days of tests of the second campaign. The sharp head decrease after the third peak of production is due to the closure of the vane at the bottom of the penstock at the end of the test.

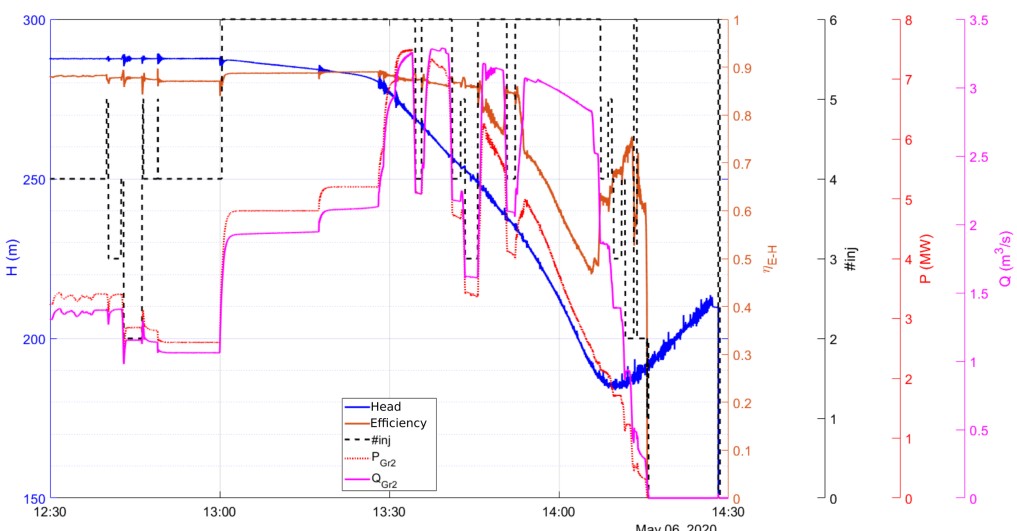

**Figure 14.** Time history of the head, the power, the efficiency, the flow discharge, and the number of opened injectors for group 2 during the third peak.

By post-processing the entire set of acquired data, the "falaise" effect can be put in evidence as shown in Figure 15, which represents the evolution of the efficiency as a function of the peripheral speed coefficient colored by the head (Figure 15a) or the power demand (Figure 15b). The emptying phase of the head race tunnel corresponds to an increase of the peripheral speed coefficient. The three different paths correspond to the three peaks carried out. The filling of the head race tunnel is always performed at minimal power. For a head higher than 250 m, the peripheral speed coefficient is below 0.52 and the efficiency is above 0.85 whatever the power. For a head around 240 m, the efficiency decreases strongly if the power demand is above 5 MW, whereas it stays above 0.8 for a lower power demand. However, when the head reaches a value between 220 m and 230 m, i.e., for a peripheral speed coefficient $k_m \approx 0.54$, the efficiency drops from 0.85 to 0.6.

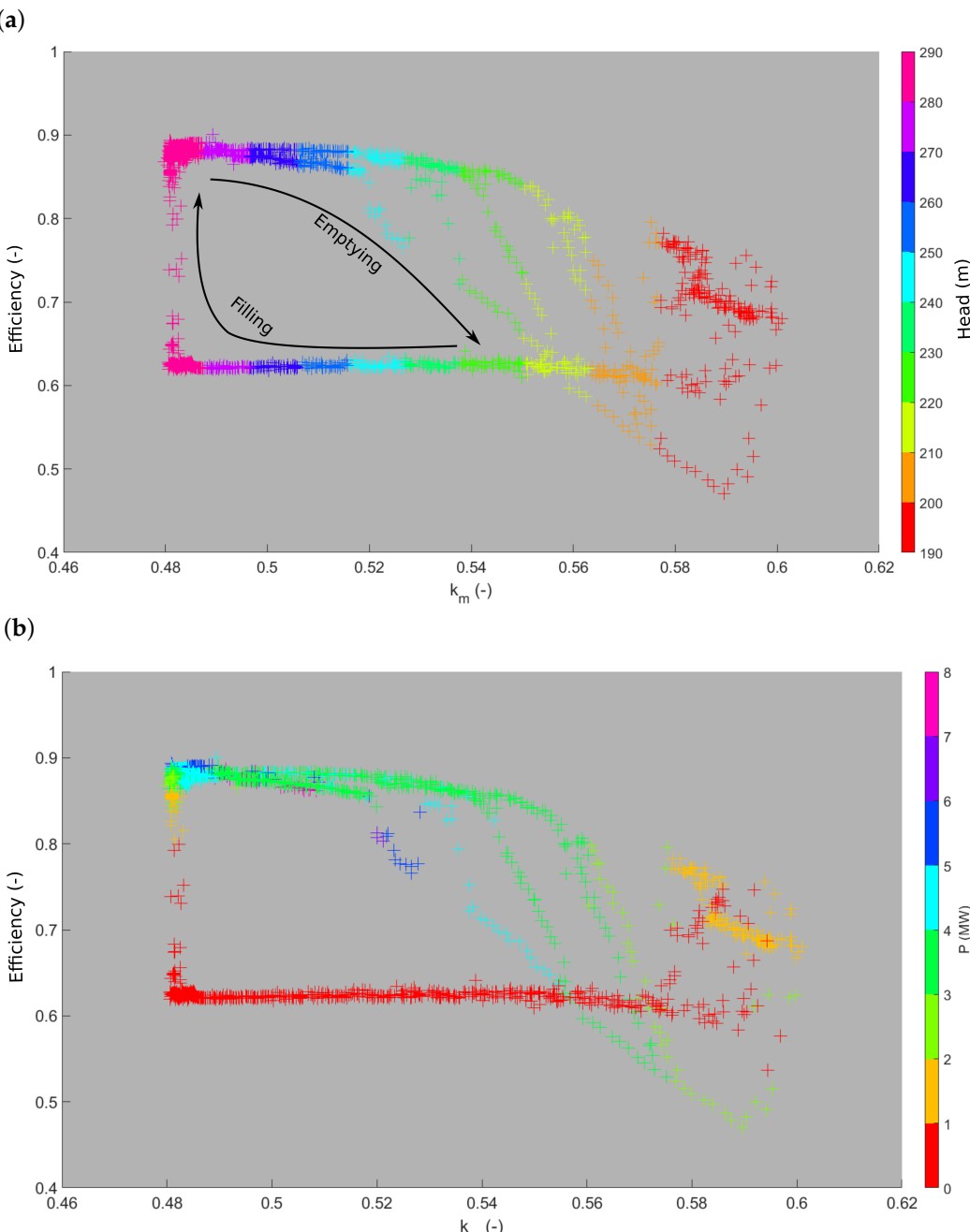

**Figure 15.** (**a**): Efficiency as a function of the peripheral speed coefficient $k_m$ colored by the head. (**b**): Idem as at the top but the symbols are colored by the power demand. The emptying and filling phases are identical for the two figures.

The 3D representation of Figure 15 provides the extended hill chart of the Pelton turbine including the region where the "falaise" effect occurs (see Figure 16).

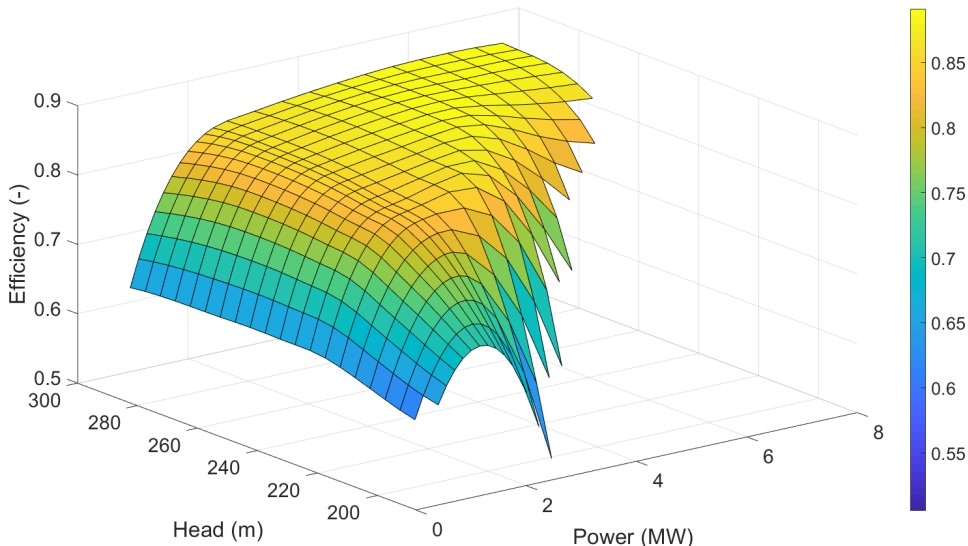

**Figure 16.** Extended hillchart of the Pelton turbine including the region where the "falaise" effect occurs.

The amplitude of the accelerometer signal along the x-direction as a function of the head is shown in Figure 17 for the three peaks. Below a head of 215 m, the amplitude increases by a factor of 4, which corresponds to the region where the efficiency drops below 0.75 (see Figure 15 at the top). The spectrogram of the signal during the third peak is plotted in Figure 18 within the addition of the time history of the head. It is noticeable that the power of all the frequencies increases when the head decreases below approximately 230 m, which corresponds to the beginning of the "falaise" effect. The constant high amplitude at 800 Hz corresponds to the bucket passage frequency.

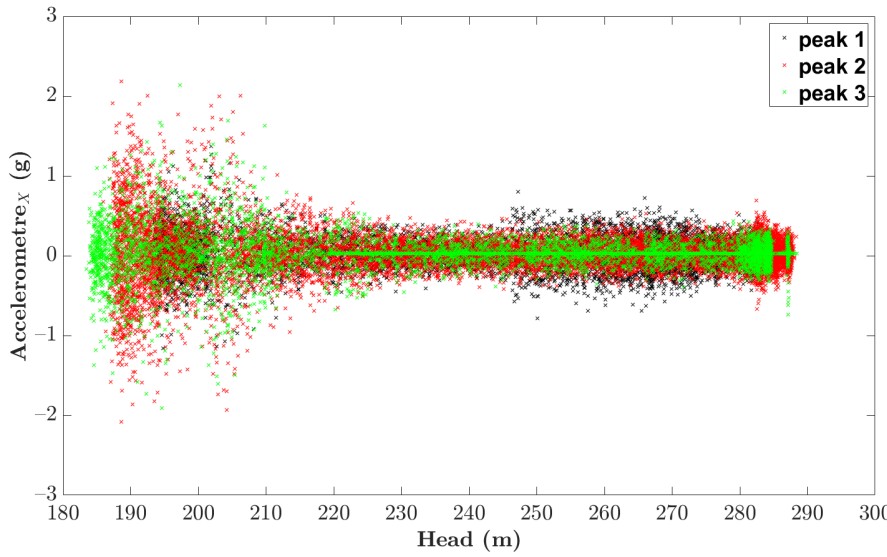

**Figure 17.** Correlation between the head and amplitude of the accelerometer along the x-direction.

In Figure 19, pictures of the free surface in the forebay tank at four different instants during the third peak are shown. The first picture shows the free surface at the beginning of the peak, when the forebay tank is fulfilled. The second picture corresponds to a forebay tank partially emptied. The third picture shows the free surface when it passes below the top of the head race tunnel and the last picture when the free surface passes through the middle of the head race tunnel. On the last picture, one can observe the flow coming from the settling basin through the gates at the bottom left close to the entrance of the head

race tunnel. These pictures do not put in evidence any vortices that develop from the free surface during the emptying phase.

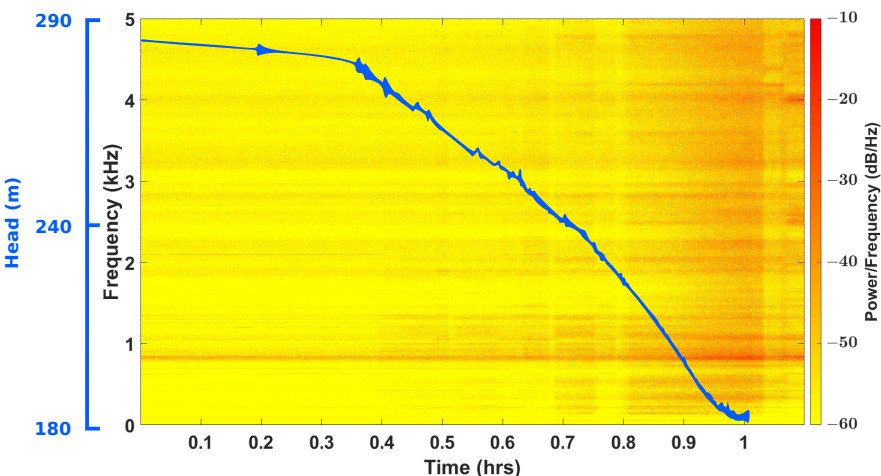

**Figure 18.** Spectrogram of the accelerometer along the x-direction during the third peak within the addition of the time history of the head (blue line).

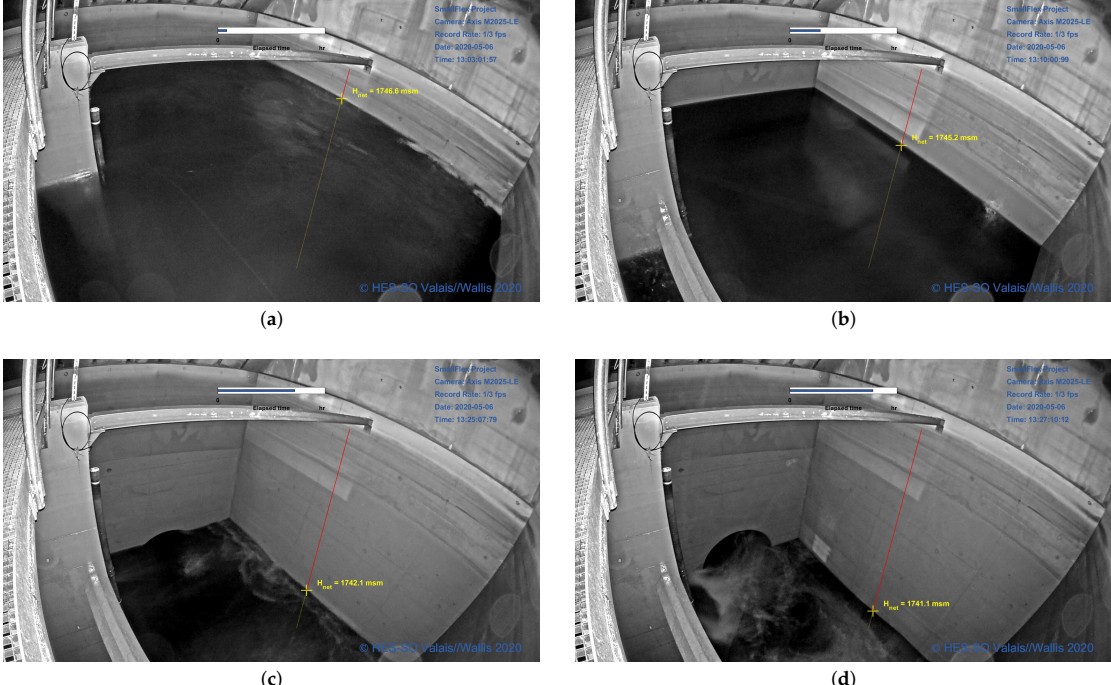

**Figure 19.** Free surface of the flow in the forebay tank at four different instants during the third peak. The red line and the yellow cross refer to the altitude of the free surface. (**a**) Free surface at the beginning of the peak, when the forebay tank is fulfilled, (**b**) free surface when the forebay tank is partially emptied, (**c**) free surface when it passes below the top of the head race tunnel, and (**d**) free surface passes when it passes through the middle of the head race tunnel.

## 6. Discussion

The on-site measurements validate the preliminary works done to plan the tests. The absence of a vortex-induced air entrainment during the emptying of the forebay tank confirms the results observed at a model scale. This observation indicates that scaling effects due to too-low Reynolds and Weber numbers seem to affect only the volume of air

entrained as shown in [24] and not the phenomena at the origin of the air entrainment, i.e., the vortex. Compared to the Stanzerthal power plant [13] characterized by a low slop of 1.5‰ along the head race tunnel to avoid the occurrence of a hydraulic jump and the risk of air entrainment downstream, the demonstrator of KWGO shows that it is also possible to use a head race tunnel with a high slope of 13% leading to the occurrence of a hydraulic jump as a storage volume without high risk of air entrainment. This is achieved thanks to a very low dimensionless flow rate $q^* = 0.012$ [19], i.e., by choosing for instance a large pipe diameter. However, this conclusion must be confirmed after a few months of operation.

The prediction of the "falaise" effect by analytical formula and the extrapolated hill chart matches with the measurements since the predictive critical peripheral speed coefficient $k_m = 0.55$ (see Section 3.2) is closed to the value around 0.54 observed on the prototype and described in the literature [20,21]. Consequently, the analytical formula can be used with confidence to estimate the occurrence of the "falaise" effect on other power plants. Beyond the efficiency drop, the prediction of the occurrence of the "falaise" effect is also necessary to avoid the increase in the vibration level by at least one order of magnitude over a wide range of frequency. Such an increase is often responsible for a premature fatigue of the components and therefore a higher risk of failure. Consequently, it can be suggested that the emptying of the head race tunnel should be limited slightly above the head at which the "falaise" effect occurs in order to keep a margin. The margin must be discussed with the operator of the power plant to keep in consideration other possible factors. In the present case, the lowest head considered should be 230 m, i.e., a head for which the efficiency is still above 0.7.

Furthermore, the tests confirm the possibility to provide a primary control service since the required power of $\Delta P = 2 \times \pm 0.75$ MW over 15 min, i.e., providing an energy of 0.375 MWh, since for instance during the third peak the average energy produced over 15 min is close to 1.2 MWh.

Finally, the hill chart of the Pelton turbine has been extended to a low head, which allows, in combination with a prediction of the inflow, to define with confidence new production plans including peaks of power over different time periods in order to provide primary control services or to adjust the production to the market price of electricity.

## 7. Conclusions

The paper presents the preparative works and the results on the on-site measurements that have been carried out to assess the capacity of a small run-of-river power plant to be operated by emptying the first third upper part of the head race tunnel.

Experiments on a model test in a laboratory have been performed to rule the risk of air entrainment out during the emptying of the head race tunnel. Moreover, the estimation of the head at which the "falaise" effect should occur has been done by analyzing the extrapolated turbine characteristic used by the digital clone and by using analytical formula. Based on these results, a minimum water level in the head race tunnel has been determined for the on-site tests including the occurrence of an emergency shutdown. Furthermore, dedicated maps of the Pelton turbine characteristics have been derived to plan production programs including the emptying of the head race tunnel without losing water as required by FMV, the owner of the power plant.

In parallel, the available power/energy for primary control services has been assessed considering both the Swissgrid requirements and the stability of the PID controller. The simulations shows that a reserve power of 1.5 MW is available throughout the year, without a risk of increasing the fatigue of the penstock.

Over two days, on-site tests including three peaks of production with the emptying of the head race tunnel were successfully achieved. Thanks to dedicated instrumentation and the monitoring via the Hydro-Clone® digital twin, several data were acquired regarding the behavior of the free surface in the forebay tank, the performances of the Pelton turbines operated at low head, and the vibration level. The on-site tests confirmed the occurrence of

the "falaise" effect around 220 m, with a sharp drop in efficiency followed by an increase in the vibration level by at least one order of magnitude.

These results could be used by the operator of the power plant to calculate the possible energy and economic gains allowed by the use of this additional storage volume. In addition, a lower limit on the head could be specified to prevent an excessive risk of fatigue. By combining this technical study with studies considering the environmental impact of the hydropeaking in the alluvial area and the economic benefits, new services requiring flexibility could be proposed in the future.

The methodology and works done in the framework of the SmallFlex project could be transposed to other small hydropower plants.

**Author Contributions:** Methodology, A.G., V.H., M.D., C.N., and C.M.-A.; validation, J.D., A.G., V.H., M.D., and C.N.; formal analysis, J.D., A.G., V.H., and M.D.; investigation, J.D., A.G., V.H., and M.D.; resources, S.C.; writing—original draft preparation, J.D. and M.D.; writing—review and editing, J.D. and M.D.; project administration, C.M.-A.; funding acquisition, C.N., S.C., and C.M.-A. All authors have read and agreed to the published version of the manuscript.

**Funding:** This research was funded by SFOE, grant number SI/501636-01 and by FMV.

**Institutional Review Board Statement:** Not applicable.

**Informed Consent Statement:** Not applicable.

**Data Availability Statement:** The final report of the SmallFlex project is available online: https: //www.aramis.admin.ch/Texte/?ProjectID=40717 (accessed on 7 July 2021).

**Acknowledgments:** The project was carried out in the framework of the Swiss Competence Center for Energy Research for the Supply of Electricity, SCCER-SoE. The authors would like to thank EAWAG, PVE, WSL, and Hydro Exploitation SA for their collaboration and support.

**Conflicts of Interest:** The authors declare no conflict of interest.

**Abbreviations**

The following abbreviations are used in this manuscript:

| | |
|---|---|
| KWGO | Kraftwerk Gletsch Oberwald |
| HPP | Hydro Power Plant |

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
