# Peer review of "Enhanced Operational Flexibility of a Small Run-of-River Hydropower Plant"

_water, doi:10.3390/w13141897_

Round 1
Reviewer 1 Report
Some additional pointers and suggestions are placed in the attachemnent

Author Response
The responds to the reviewer's comments are attached in the PDF file.
Reviewer 2 Report
I had the opportunity to read the paper titled:
Enhanced operational flexibility of a small run-of-river hydropower plant by Decaix et al. The paper deals only with the technical assessments of the new flexibility offered by emptying the settling basin, the forebay tank and the third upper part of the head race tunnel in a small hydropower plant in Switzerland. The study covers an interesting issue, i.e. the electricity production from renewable energies, particularly at a time of growing global interest in increasing the part of renewable energy in the current energy mix and in the meantime at stopping nuclear power plants. By combining investigations performed on a laboratory test rig, numerical simulations and on-site measurements, it gathers very valuable information on the flexibility can be achieved by using hydropower plants that have storage capacity allowing to shift in time the production of electricity to ensure the production-demand balance on the electricity grid. This information can be useful to (small) hydraulic power plants owners who can enhance the production of electricity and plan production programs without too high environmental and economic costs. However, in the absence of a well-supplied international literature, the presentation of the paper and its content seems to me too much like an expert report than a scientific article. This feeling is further supported by the lack of clearly defined hypotheses for this study as well as that of the Discussion section.
In my opinion, this article requires extensive amendments before any consideration. I suggest a revised version that focuses on the relevant aspects/information that the authors want to bring to the attention of the public/hydropower plant owners and scientists while leaving the context too Swiss through a well-documented bibliography and discussed results, which compare their findings with other studies.
As specific comments, there are:
Line 1-5 This introductive part of the abstract can be improved. It seems to be too much based on a local/national problem.
Line 5 What is meant by FMV? (see line 5, 32-33, 47,212,276,306) . Ditto for PVE, WSL, EAWAG and EPFL.
Line 17-44. Introduction too closed and limited to only the flexibility of run-of-river power plant production.
Line 40-44 This presentation/style reminds us that we are in the presence of an expert report.
Line 53 Please make the style uniform “m³/s” with “0.1s-1”.
Line “and [15]” please change to “and Novotny 15]”.
Line 80-82 On what scientific basis has the external expert certified the absence of risk at least among several cycles of emptying and filling operations? This needs to be discussed.
Line 66 in Fig. 1 What is meant by altitude 1450 a.s.l.? see also L and D. Abbreviations needs to be explained in the caption.
Line 108 What is meant by HPP?
Line 91-108 For the protocol on investigation of the risk of air entrainment, the authors do not refer to any bibliographic references.
Line 112 "Following figure 16.2 in [17]", please improve this style.
Line 171-177 The number of on-site tests seems to me insufficient to draw reliable implications.
Line 179-188 Abbreviations should be explained in the caption of the figure 9 for easy reading.
Pages 9-13 line 222-264 Two days of testing do not seem to me to be enough to draw any conclusions.
Line 242-243 Insufficient interpretation, this further justifies the inclusion of the section Discussion of the results obtained. Here also a reference should be provided.
Line 222-254 The results are very interesting. However, they are not discussed. The discussion section is completely absent.
Line 254-260 These should be summarised in the legend for better understanding and easy reading of Figure 18.
Figure 18 Pictures difficult to follow because the caption does not explain anything. Readers should be able to understand the figure without resorting to the main text.
The bibliographical reference consulted is too poor (at most 5 articles published in specialised journals, pages 15-16) for this paper to be considered for publication in a journal as important as Water.
Author Response
The answers to the reviewer's comments are provided in the attached PDF file.

Round 2
Reviewer 2 Report
I am positive to this MS, which seems unique in its technical assessments of the new flexibility offered by emptying the settling basin, the forebay tank and the third upper part of the head race tunnel in a small hydropower plant in perspective to improve the electricity production from renewable energies. However, the importance and usefulness of its findings are obscured by the lack of a robust, critical, balanced discussion and comparison of the observed performance with other relevant works in the same fields of study as well as the lack of use of a well documented literature.
I am not entirely satisfied with the amendments made. In my opinion, the paper should not be published but it would be better if the authors resubmitted it as a "short communication" or "technical report" and not as a regular article.
Author Response
The response to the editor's comments are attached in the PDF file

Round 3
Reviewer 2 Report
I must congratulate the authors for the effort done in reviewing the manuscript as some of the changes included in this version in the sections of the Investigation of the risk of air entrainment and the Discussion now improved the paper.
I also thank the authors to have considered some questions/remarks/suggestions raised during the review process. Their adaptations and explanations seem sufficient to allow a final acceptation for publication.